# A scoping review on the association between early childhood caries and life on land: The Sustainable Development Goal 15

Morénike Oluwátóyìn Foláyan[1,2,3,4]*, Robert J. Schroth[1,5], Duangporn Duangthip[1,6], Ola B. Al-Batayneh[1,7,8], Jorma I. Virtanen[1,9], Ivy Guofang Sun[1,6], Arheiam Arheiam[1,10], Carlos A. Feldens[1,11], Maha El Tantawi[1,3,4,12]

1 Early Childhood Caries Advocacy Group, University of Manitoba, Winnipeg, Canada, 2 Department of Child Dental Health, Obafemi Awolowo University, Ile-Ife, Nigeria, 3 Oral Health Initiative, Nigerian Institute of Medical Research, Yaba, Lagos, Nigeria, 4 Africa Oral Health Network, Alexandria University, Alexandria, Egypt, 5 Dr. Gerald Niznick College of Dentistry, Rady Faculty of Health Sciences, University of Manitoba, Winnipeg, Canada, 6 Faculty of Dentistry, The University of Hong Kong, Hong Kong SAR, China, 7 Department of Orthodontics, Pediatric and Community Dentistry, College of Dental Medicine, University of Sharjah, Sharjah, United Arab Emirates, 8 Preventive Dentistry Department, Jordan University of Science & Technology, Irbid, Jordan, 9 Faculty of Medicine, University of Bergen, Bergen, Norway, 10 Department of Dental Public Health, Faculty of Dentistry, University of Benghazi, Benghazi, Libya, 11 Department of Pediatric Dentistry, Universidade Luterana Do Brasil, Canoas, Brazil, 12 Department of Pediatric Dentistry and Dental Public Health, Faculty of Dentistry, Alexandria University, Alexandria, Egypt

* toyinukpong@yahoo.co.uk

**Data Availability Statement:** All underlying data are in the paper.

## Abstract

### Background

The Sustainable Development Goal 15 (SDG15) deals with protecting, restoring, and promoting the sustainable use of terrestrial ecosystems, sustainably managing forests, halting and reversing land degradation, combating desertification and halting biodiversity loss. The purpose of this scoping review was to map the current evidence on the association between SDG 15 and Early Childhood Caries (ECC).

### Methods

This scoping review was reported in accordance with the preferred reporting items for systematic reviews and meta-analyses extension for scoping reviews (PRISMA-ScR) guidelines. Formal literature searches were conducted in PubMed, Web of Science, and Scopus in March 2023 using key search terms. Studies with the criteria (in English, with full text available, addressing component of life on land, focusing on dental caries in humans, with results that can be extrapolated to control ECC in children less than 6 years of age) were included. Retrieved papers were summarised and a conceptual framework developed regarding the postulated link between SDG15 and ECC.

### Results

Two publications met the inclusion criteria. Both publications were ecological studies relating environmental findings to aggregated health data at the area level. One study concluded

**Funding:** The author(s) received no specific funding for this work.

**Competing interests:** The authors have declared that no competing interests exist.

that the eco-hydrogeological environment was associated with human health, including caries. The other reported that excessive calcium was associated with the presence of compounds increasing groundwater acidity that had an impact on human health, including caries. The two ecological studies were linked to SDG 15.1. It is also plausible that SDG 15.2 and SDG 15.3 may reduce the risk for food insecurity, unemployment, gender inequality, zoonotic infections, conflict and migration; while SDG 15.4 may improve access to medicinal plants such as anticariogenic chewing sticks and reduction in the consumption of cariogenic diets.

## Conclusions

There are currently no studies to support an association between ECC and SDG15 although there are multiple plausible pathways for such an association that can be explored. There is also the possibility of synergistic actions between the elements of soil, water and air in ways that differentially affect the risk of ECC. Studies on the direct link between the SDG15 and ECC are needed. These studies will require the use of innovative research approaches.

## Introduction

The World Health Organization's sustainable development goal 15 (SDG15) seeks to protect, restore, and promote the sustainable use of terrestrial ecosystems, sustainably manage forests, halt and reverse land degradation, combat desertification, and halt biodiversity loss [1]. It addresses water, land, food, air, and soil related issues [2]. The SDG15 is important because human life depends on the earth for sustenance and livelihoods. The land, directly or indirectly, provides clean water and air, food and essential nutrients, medicines and medicinal compounds, energy, livelihoods, and spiritual, cultural, and recreational enrichment [3]. Forests cover 30% of the Earth's surface and provide vital habitat for millions of species and are an important source of clean air and water [4].

Degradation, desertification and biodiversity loss disproportionately affect poor communities [5]. Environmental degradation, caused by greenhouse and carbon dioxide emissions [6], negatively affects people's health, reduces life expectancy, increases health expenses [7, 8], and adversely affects economic activities [9, 10]. Land degradation and desert expansion lead to meteorological (hydrological imbalances that adversely affect land resource production systems) [11], hydrological (surface and/or subsurface water supply is affected), socioeconomic (demand of commodities exceeds supplies due to water scarcity), and agricultural (shortage of precipitation impinges on crop growth due to soil moisture drought) negative effects [12]. These result in decreased food production, water shortages, sanitation and hygiene challenges, population migration to move to more hospitable areas, and problems with air and soil quality all of which, have negative impact on health [2, 13]. Biodiversity loss also increases the rate of wildlife–human contacts, thereby increasing the risks for infectious diseases [14, 15].

The indirect complex pathways for the negative impact of land degradation, desertification, and biodiversity loss on general health—food production, freshwater access, and ecosystem resources [2]–may be associated with increased risk of early childhood caries (ECC) through multiple pathways. ECC is the presence of cavitated or non-cavitated caries lesions involving the primary dentition in children < 72-months of age [16]. It is caused by multi-layered risk factors that operate at individual, family, and society level. Land degradation may cause food

and water insecurity, unemployment, gender inequality, conflict, and migration, all of which are risk factors for ECC [17–20]. Emerging and re-emerging infectious diseases resulting from wildlife–human contacts, population and household crowding and poor sanitation, may also have detrimental oral health effects [21–23]. For example, higher risk of ECC was reported in relation to the COVID-19 pandemic [21, 24]; and disruption of the bio-environment increases the risk for oral diseases [25].

These contributory pathways of the SDG 15 to the risk for ECC is of concern as the current global burden of ECC is high. Caries in the primary dentition affects 514 million children globally [26] with prevalence ranging from 82% in the Oceania to 52% in Asia, 48% in the Americans, 43% in Europe and 30% in Africa [27]. The regions and countries worst affected by food insecurity [28], water insecurity [29], unemployment [30], gender inequality [31], conflict [32], and migration [33]–Asia and Africa—are regions with high but not the worst prevalence of ECC globally [27]. These regions in need of extensive intensive interventions for SDG 15 may benefit from a reduction in the prevalence of ECC if strategic actions for ECC control are also linked and monitored alongside efforts to address the SDG 15. Though SDG 15 and ECC may be linked, the empirical evidence for this is needed to enable strategic plans for interventions. The aim of this scoping review was to map the current evidence on the association between SDG 15 and ECC. It specifically explored if there was an association between land degradation, desertification, and biodiversity loss and ECC.

## Methods

A systematic search was performed to identify literature on the association between life on land related factors and ECC. The Scoping Review was reported in accordance with the Preferred Reporting Items for Systematic Reviews and Meta-Analyses Extension for Scoping Reviews (PRISMA-ScR) guidelines [34, 35]. The protocol on how to proceed through the search strategies, eligibility assessment, selection of studies and data extraction was planned and followed but the study was not registered [36].

### Research question

This review was guided by the question: What is the existing evidence on the association between life on land and ECC?

### Search strategy

Formal literature searches were conducted in PubMed, Web of Science, and Scopus in March 2023. Search terms and strategies used in each database are reported in S1 File. The three databases searched are prominent databases that serve as repositories for high quality manuscripts in dentistry.

### Eligibility criteria

Published literature obtained through database searches was exported to the reference management software Mendeley, where duplicates were removed.

### Inclusion criteria

Studies were included if they reported information on the association between life on land related factors outlined in SDG 15 and ECC before March 2023. These factors included conserving and restoring terrestrial and freshwater ecosystems; ending deforestation and restoring degraded forests, ending desertification and restoring degraded land, ensuring conservation of

mountain ecosystems, protecting biodiversity and natural habitats, promoting access to genetic resources and fair sharing of benefits, eliminating poaching and trafficking of protected species, preventing invasive alien species on land and in water ecosystems, integrating ecosystem and biodiversity in governmental planning, increasing financial resources to conserve and sustainably use ecosystem and biodiversity, and financing and incentivizing sustainable forest management.

Studies were included if they reported information on caries in the primary dentition in children <6 years of age, in keeping with the case definition for ECC, or allowed extrapolation to ECC. Only those publications for which full texts were available and papers written in English were included. No restrictions were imposed based on date of publication or the type of study. Studies that show a risk for caries/ECC resulting from wildlife–human contacts were also eligible for inclusion.

### Exclusion criteria

Studies on SDG 15 linked directly with other SDGs were excluded to eliminate the risk of overlaps in the research team's findings between the link ECC and each SDG. These include those linked with SDG 2 (food production), SDG 6 (water scarcity, sanitation and hygiene), SDG 10 (migration resulting degradation, desertification and biodiversity loss) and SDG 13 (greenhouse and carbon dioxide emissions).

### Data charting

Title and abstract screening were conducted by two independent reviewers (MET and MOF), guided by eligibility criteria. No authors or institutions were contacted to identify additional sources. Full-text reviews of the remaining publications were then completed independently by the same two researchers (MET and MOF) and reference lists of potentially relevant publications were manually searched. Uncertainty regarding whether publications met the inclusion criteria was resolved through discussion among the two independent reviewers.

The name of the author, the publication year of the manuscript, study location, study design, study aim, data collection methods, and main findings were extracted from the studies included in this review. A descriptive analysis of the extracted data was conducted. Fig 1 shows the flowchart of study selection process.

The summarized data were shared with one expert for his review (MA). Publications were retained only when there was consensus between the experts and the earlier two reviewers. The consensus document was then shared with members of the Early Childhood Caries Advocacy Group (www.eccag.org) for confirmation.

## Results

The initial searches yielded 114 publications. After removing duplicates, 111 publications remained. Following the scanning titles and abstracts, 108 papers were excluded. One additional study was excluded after full text review because it addressed caries in children beyond 6 years of age. Overall, two papers were included in this scoping review [37, 38] (Fig 1). Both included publications were determined to be ecological studies relating environmental findings at area level to aggregated health data in the same area. Both articles were published in non-dental journals (Table 1).

The first study was published in 2004 by a research team from China and the United Kingdom. The publication focused on modelling the leaching, transferring and enrichment of elements in a hydrogeochemical system in northern China and related caries levels to changes occurring in elements sampled from mountains, highlands, shallow aquifers, and deep aquifers

**Identification of studies via databases**

Identification

Records identified from*: (n=114)
 Databases
 PubMed = 19
 Scopus= 8
 Web of Science= 87

Records removed before the screening: (n=3)
 Duplicate records removed (n = 3)
 Records marked as ineligible by automation tools (n = 0)
 Records removed for other reasons (n = 0)

Screening

Records screened
(n = 111)

Records excluded**
(n = 109)

Reports sought for retrieval
(n = 2)

Reports not retrieved
(n = 0)

Reports assessed for eligibility
(n =2)

Reports excluded: (n=0)

Included

Studies included in review
(n = 2)

**Fig 1. Study flowchart showing the flow of studies from retrieval to the final included studies.**

[37]. The authors concluded that the eco-hydrogeological environment was associated with human health, including caries.

The second study was published in 2021 by a group of researchers from Pakistan and Finland. The study assessed the changes in groundwater quality due to land use and land cover in Peshawar, Pakistan over a period of eight years. Excessive calcium, indicative of water hardness, was associated with the presence of compounds increasing groundwater acidity, which was associated with an impact on human health, including caries [38].

No empirical study was retrieved on the direct relationship between SDG 15-related indicators and ECC. The two ecological studies included in this review revealed that the

**Table 1. Summary of studies included in the scoping reviews.**

| Author (Publication year) | Study site | Study design | Study objective | Findings |
|---|---|---|---|---|
| Cao et al. (2004) [37] | China | Ecological study | Identifying the importance and variety of links between hydrogeology, ecology and human health, and address the significance of geological environment to human beings. | Higher prevalence of dental caries was reported where there was a lower concentration of fluoride. |
| Ahmad et al. (2021) [38] | Pakistan | Ecological study | Evaluation of temporal changes of groundwater quality from 2012 to 2019, its relation to land use/landcover, and its impact on residents of Peshawar. | An increase in dental caries was associated with the high calcium and magnesium concentration in the groundwater samples. |

conservation, restoration and sustainable use of terrestrial and inland freshwater ecosystems and their services, in particular forests, wetlands, mountains and drylands (SDG 15.1) may affect the ionic content of the ecosystems, which may have implications for children's caries risk. It is also plausible that the sustainable management of all types of forests (SDG 15.2) and combating desertification and restoring degraded land and soil (SDG 15.3) may reduce the risk for food insecurity, unemployment, gender inequality, zoonotic infections, conflict and migration [39] which are important to reduce the risk for ECC [15, 17–20]. Also, not conserving mountain ecosystems (SDG 15.4) may damage the bridge that links the natural environment and human wellbeing [40]; lead to the loss of biodiversity and reduce access to medicinal plants, and promote caries-inducing diets [41]. There are, however, no studies yet available to support these possible associations. The absence of empirical studies does not preclude the possibility of a link. The probable conceptual framework between SDG 15 and ECC is shown in the Fig 2.

## Discussion

This present scoping review reveals that there is no primary research-derived evidence on the association between SDG 15 and ECC based upon the English literature search. However, the two ecological studies provide suggestive evidence, howbeit weak, that can be extrapolated on the associations between SDG 15 and ECC. Several important findings emerge. First, land use, the degradation of soil, and hydrological conditions play important roles in influencing the composition of dissolved organic matter in the aquatic ecosystem and reservoir water quality [42, 43]. Land use also has an impact on soil leaching and soil acidity [44, 45], which has an impact on its concentration and bioavailability of trace elements [46]. The margin of safety between beneficial and harmful levels of trace elements in the environment is narrow: deficiencies are linked to plant nutritional problems while a build-up of trace elements may have a negative effect on health [46]. For example, high concentrations of natural fluoride in groundwater and soils may cause enamel defects, delay in tooth eruption and endemic skeletal and dental fluorosis [47–52]. High fluoride concentration also leads to poor soil mineral fertility and poor protein production where soil has a low capacity to produce legumes, beef, and mutton and has been growing starchy grains and fattening live stocks. All this may be linked to caries [44].

Second, soil, water, and air are sources of multiple trace element that can affect oral health. Several trace elements are linked to the risk of caries [53] either as cariostatic such as molybdenum, strontium, vanadium, fluoride and lithium or cariogenic including selenium, cadmium, manganese, copper, lead and zinc [54]. Vanadium is found naturally in soil, water, air, continental dust, sea aerosol, and volcanic emissions [55]. Strontium is found everywhere in the environment [44]. Lithium is found in drinking water and other environmental sources [56] while copper is found in water and soil [57]. Molybdenum is found in water [58] while

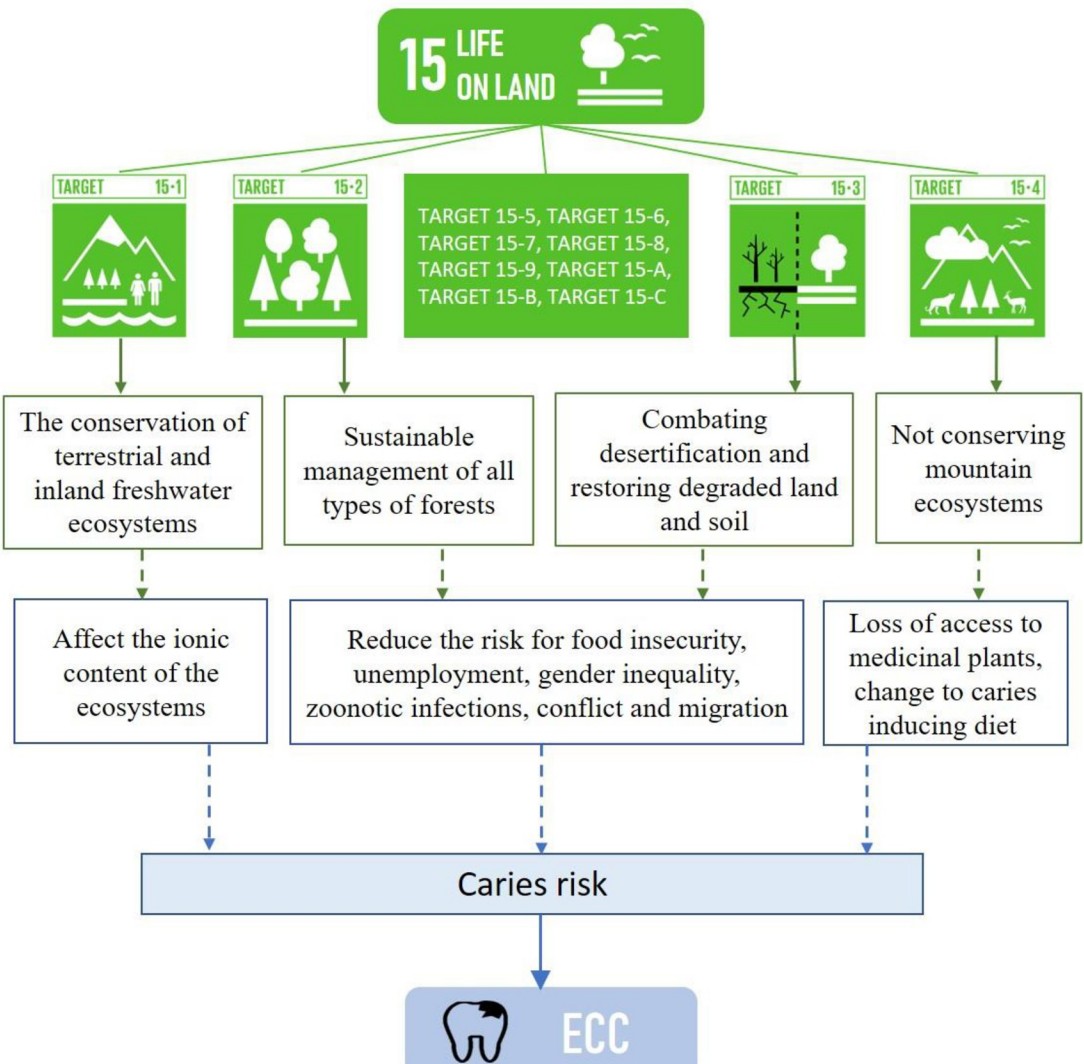

**Fig 2. The conceptual framework between SDG 15 and ECC.** (The targets of SDG 15 are available at https://www.globalgoals.org/goals/15-life-on-land/).

selenium is found in minute amounts in all materials of the earth's crust and water-soluble selenate in well-aerated alkaline soils [59]. Lead is found in earth crust, air, water, and soil [60], while manganese and zinc are found in soil [61, 62], and cadmium in soil and ground water [63]. Studies on the link between the presence of these elements in soil, water, or air in the context of sustainable terrestrial ecosystems and ECC are needed.

It is suggested that sustainable life on land may affect the risk of ECC in multiple ways. One way is through the chemical-biological-physiological pathway. For example, trace elements may increase the rate of post-eruptive maturation (e.g., fluoride), remineralization of incipient lesions (e.g., fluoride, zinc, strontium), interfere with plaque microorganisms and the ability to produce acid (e.g., zinc, calcium, fluoride, lead, copper, nickel, gold, silver, mercury), reduce the loss of calcium from the tooth (e.g., strontium), or change the tooth morphology and micro-crystal structures (e.g., fluoride, zinc, selenium, lead) [64–75]. These properties result from incorporating trace metals into the apatite microcrystals of enamel or dentine, altering

their physical properties, especially their solubility and susceptibility to degradation by acid [75]. These effects are exerted during the development of the tooth or at the post-developmental period. Fluoride exerts a developmental and post-developmental caries protective effect [76]. The post-developmental topical effect of fluoride is through the constant maintenance of fluoride in the oral cavity to interfere with the process of caries lesion development [77]. Cadmium on the other hand, is caries-preventing during the developmental period, but caries-inducing during the post-developmental period [78].

Another way is the cultural-social-economic pathway. This is an under researched area that still needs to be explored. Little is known about the ECC risk in communities that have to make adaptation choices to the environmental challenges they face as a result of exposure to desertification, land degradation, and biodiversity loss [79]. These adaptation choices may increase or decrease the risk of ECC through dietary habits, oral hygiene behaviours, or psychosocial profile [80, 81]. For example, the loss of biodiversity and medicinal plants may reduce access of people in remote areas to chewing sticks that are anticariogenic [82–84] in the absence of access to fluoridated toothpastes. Also, desertification and the loss of mountain bioecosystems increases the risk of loss of nature stewardship role and employment opportunities for women who are the carers for children [85]. Low maternal income increases the risk of ECC [86]. These wider socio-economic changes may affect the oral health behaviours of families and their young children. However, this remains an assumption that needs to be explored.

Third, the distribution of trace elements in soil in different regions and on different scales has spatial heterogeneity [87]. This spatial heterogeneity of soil, water, and air may contribute to the geospatial variability in the epidemiological profile of ECC. Early research seemed to link the pattern of soil fertility to caries prevalence [43]. Soil fertility is linked to ground water connection [88]. The ground waters in areas of Africa, Asia, the Middle East, Southern Europe, and the Southern USA contain high concentrations of fluoride, well above the 'optimum level' of 1 ppm; while the level is too low to prevent and control caries in some other locations [89]. The natural presence of fluoride in water and soil (soils act as a sink rather than a source of fluoride [90]) allows for its constant ingestion, saliva secretion and continuous bathing of the teeth by fluoride-enriched saliva [91]. This is the most plausible mechanism of action of the earth and the soil on caries. Fluoride in water is effective at reducing caries levels in both primary and permanent dentition in children by providing a constant exposure to fluoride ions in the oral cavity [92]. Supra-optimal levels have been linked to dental fluorosis, and severe dental fluorosis is a risk factor for caries [93].

We observed that the relationships between land and ECC seem to be complex and a deeper understanding of the inter-linkages between the oral health and life on land is needed. Ecological studies may provide preliminary evidence suggesting the plausibility of these links. Understanding these linkages will facilitate the integration of ECC prevention and management into policies and programs that aim to protect the environment. "Health in All" policies address risks to the environment and to health within the sustainable development framework [94]. The "Health in All" policies is an approach to the formulation of public policies across sectors that systematically consider the health implications of decisions, seeks synergies, and avoids harmful health impacts to improve population health and health equity [95]. The Health in All is a complementary approach to the SDG that promotes linking the SDGs [96] and monitoring the impact of policies, plans and programmes that promote the SDGs on health using the Health Impact Assessment tools [97]. The Health Impact Assessment tools facilitate the identification of the pathways of the SDG 15's impact on oral health by considering multiple exposures and complexities, interdependencies, and uncertainties of the real world [98]. Establishing a link between SDG 15 and ECC supports advocates in the field of child and oral health by taking actions to reduce the degradation of natural habitats and halt the loss of

biodiversity (SDG 15.5, 15.9, 15.a, 15.b. 15.c). Thus, policies reducing land degradation, desertification and diversity loss could, at the same time, positively impact oral health and prevent ECC.

The scoping review has a few limitations. These include the exclusion of manuscripts not written in English and grey literatures thereby limiting the scope of coverage. Despite this limitation, this scoping review highlights for the first time, the possible relationship between SDG15 and ECC. We recommend future research to clarify the link between life on land and ECC across the hierarchy of evidence spectrum [99], including lab studies comparing trace elements in carious versus non-carious teeth, cross sectional studies assessing the impact of environmental stresses caused by natural or man-made disasters on land productivity and people's livelihoods reflected on general and oral health, clinical trials assessing the effect of medicinal plants on curing oral diseases and improving oral health in addition to studies using techniques such as causal inference [100] to dissect the mechanism through which challenges to life on land impact oral health in general and ECC in particular. It is also important to involve various stakeholder in this future research by acknowledging the chemico-biological and socio-cultural effects of life on land on oral health by including researchers from life sciences and humanities to provide a holistic overview of the association. Representation of participants from settings of different geographic and economic backgrounds is also important.

## Conclusion

Although we found very limited empirical evidence on the link between the SDG15 and ECC, there is the plausibility of such associations through complex socioeconomic and health behaviour pathways that are yet to be fully understood. There is also the possibility of synergistic actions between the elements of soil, water and air in ways that differentially affect the risk of ECC. There is currently no study on the direct link between the SDG15 and ECC. These studies are needed and will require the use of innovative and facilitated research approaches.

## Supporting information

**S1 Checklist. Preferred Reporting Items for Systematic reviews and Meta-Analyses extension for Scoping Reviews (PRISMA-ScR) checklist.**
(DOCX)

**S1 File. Search strategy for Scopus, Pubmed and Web of Science on the links between SDG 15 and Early Childhood Caries.**
(DOCX)

## Author Contributions

**Conceptualization:** Morẹ́nikẹ́ Oluwátóyìn Foláyan, Maha El Tantawi.

**Data curation:** Morẹ́nikẹ́ Oluwátóyìn Foláyan, Maha El Tantawi.

**Formal analysis:** Morẹ́nikẹ́ Oluwátóyìn Foláyan, Maha El Tantawi.

**Investigation:** Morẹ́nikẹ́ Oluwátóyìn Foláyan.

**Methodology:** Morẹ́nikẹ́ Oluwátóyìn Foláyan, Duangporn Duangthip, Jorma I. Virtanen, Ivy Guofang Sun, Carlos A. Feldens, Maha El Tantawi.

**Project administration:** Morẹ́nikẹ́ Oluwátóyìn Foláyan, Robert J. Schroth, Maha El Tantawi.

**Supervision:** Morẹ́nikẹ́ Oluwátóyìn Foláyan, Robert J. Schroth.

**Validation:** Morénikẹ́ Oluwátóyìn Foláyan, Robert J. Schroth, Jorma I. Virtanen, Arheiam Arheiam, Carlos A. Feldens, Maha El Tantawi.

**Visualization:** Duangporn Duangthip, Ivy Guofang Sun.

**Writing – original draft:** Morénikẹ́ Oluwátóyìn Foláyan.

**Writing – review & editing:** Morénikẹ́ Oluwátóyìn Foláyan, Robert J. Schroth, Duangporn Duangthip, Ola B. Al-Batayneh, Jorma I. Virtanen, Ivy Guofang Sun, Arheiam Arheiam, Carlos A. Feldens, Maha El Tantawi.

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
