## [Decision Letter · Decision Letter 0]

17 Mar 2024

PONE-D-23-43962A scoping review on the association between early childhood caries and life on land: the Sustainable Development Goal 15PLOS ONE

Dear Dr. Foláyan,

Thank you for submitting your manuscript to PLOS ONE. After careful consideration, we feel that it has merit but does not fully meet PLOS ONE’s publication criteria as it currently stands. Therefore, we invite you to submit a revised version of the manuscript that addresses the points raised during the review process.

We look forward to receiving your revised manuscript.

Kind regards,

Hadi Ghasemi

Academic Editor

PLOS ONE

Journal Requirements:

Reviewers' comments:

Reviewer's Responses to Questions

**Comments to the Author**

1. Is the manuscript technically sound, and do the data support the conclusions?

Reviewer #1: Partly

Reviewer #2: Yes

Reviewer #3: Yes

2. Has the statistical analysis been performed appropriately and rigorously? 

Reviewer #1: N/A

Reviewer #2: Yes

Reviewer #3: Yes

3. Have the authors made all data underlying the findings in their manuscript fully available?

Reviewer #1: No

Reviewer #2: Yes

Reviewer #3: Yes

4. Is the manuscript presented in an intelligible fashion and written in standard English?

Reviewer #1: Yes

Reviewer #2: Yes

Reviewer #3: Yes

5. Review Comments to the Author

Reviewer #1: PONE-D-23-43962 A scoping review on the association between early childhood caries and life on land: the Sustainable Development Goal 15

This manuscript describes a scoping review of the evidence linking Sustainable Development Goal 15 (SDG15) and early childhood caries. Is interesting and novel.

INTRODUCTION

1. The description of Sustainable Development Goal 15 (SDG15) is appropriate, but a general description of the global prevalence and geographical distribution of ECC is lacking. Systematic reviews of the global distribution of ECC are currently available and could be used to link this data to SDG15.

2. However, one might think that the prevalence of ECC, if associated, should follow a geographical distribution, and the reviews of the global prevalence of ECC do not show this association. This should be discussed.

3. "Although there is suggestive evidence of a link between SDG 15 and ECC, there is little empirical evidence to support this possible relationship." This sentence needs a citation or reference to support who and why this association is suggested, and these references need to be explored in detail.

4. The literature review found is superficial and could be enriched by considering the key component of the research question: what is the relationship between the proposed risk factors and ECC? For this, I believe that the framework proposed by Hill around 1950 is still valid and should be considered: is there evidence of temporality? Strength? Consistency? Specificity? Biological gradient? Plausibility, Coherence? Analogy and experimental? This should be explored in detail in this exploratory study, so a table summarising whether we have evidence to check each of the Hill criteria and the reference could summarise the results

5. The sentence "High concentrations of natural fluoride in groundwater and soils may cause enamel defects, delayed tooth eruption and endemic skeletal and dental fluorosis [39-44]" needs to be balanced with the fact that caries prevalence is also lower in these areas. See Kut et al. 2016. A review of fluoride in African groundwater and local remediation methods. Groundwater for Sustainable Development 2-3, 190-212.

6. The raw data of potential papers should be included as supplementary material or in a dedicated data research repository.

Reviewer #2: The research will contribute to the literature. It is a topic that remains current and has undergone a good literature review. Being a multicentric study increases its value. The sources could have been more up-to-date in terms of the year. It is important to highlight the need for more research related to ecological studies.

Reviewer #3: The manuscript 'A scoping review on the association between early childhood caries and life on land:

the Sustainable Development Goal 15' is an excellent and need of the time study.

PRISMA-ScR guidelines are followed well.

Whole manuscript is written in standard English.

Results are meticulously drawn and are well interpreted.

Conclusion and need of further studies are nicely mentioned.

6. PLOS authors have the option to publish the peer review history of their article (what does this mean?). If published, this will include your full peer review and any attached files.

Reviewer #1: No

Reviewer #2: No

Reviewer #3: **Yes: **Prof. Divya S Sharma

---

## [Author Response · Author response to Decision Letter 0]

1 May 2024

A scoping review on the association between early childhood caries and life on land: the Sustainable Development Goal 15

PONE-D-23-43962

Date of revision: 18th March 2024

We authors would like to thank the reviewers for their critical review of the manuscript. The points raised has helped strengthen the paper. The positive feedback is encouraging. Please find below, our point-by-point responses to the issues raised by the reviewers.

Reviewer #1

This manuscript describes a scoping review of the evidence linking Sustainable Development Goal 15 (SDG15) and early childhood caries. Is interesting and novel.

Response: Thanks for the positive feedback.

INTRODUCTION

1. The description of Sustainable Development Goal 15 (SDG15) is appropriate, but a general description of the global prevalence and geographical distribution of ECC is lacking. Systematic reviews of the global distribution of ECC are currently available and could be used to link this data to SDG15.

Response: Thanks for highlighting this gap in the introduction. We have revised the introduction and wrote in the last paragraph: These contributory pathways of the SDG 15 to the risk for ECC is of concern as the current global burden of ECC is high. Caries in the primary dentition affects 514 million children globally [26] with prevalence ranging from 82% in the Oceania to 52% in Asia, 48% in the Americans, 43% in Europe and 30% in Africa [27]. 

2. However, one might think that the prevalence of ECC, if associated, should follow a geographical distribution, and the reviews of the global prevalence of ECC do not show this association. This should be discussed.

Response: You are right. We noticed this and we wrote: The regions and countries worst affected by food insecurity [28], water insecurity [29], unemployment [30], gender inequality [31], conflict [32], and migration [33] – Asia and Africa - are regions with high but not the worst prevalence of ECC globally [27]. These regions needing extensive intensive interventions for SDG 15 may benefit from a reduction in the prevalence of ECC if strategic actions for ECC control is also linked and monitored alongside efforts to address the SDG 15. 

3. "Although there is suggestive evidence of a link between SDG 15 and ECC, there is little empirical evidence to support this possible relationship." This sentence needs a citation or reference to support who and why this association is suggested, and these references need to be explored in detail.

Response: Thanks for raising this. We have revised the statement and wrote: Though SDG 15 and ECC may be linked, the empirical evidence for this is needed to be able to make strategic plans for interventions. 

4. The literature review found is superficial and could be enriched by considering the key component of the research question: what is the relationship between the proposed risk factors and ECC? For this, I believe that the framework proposed by Hill around 1950 is still valid and should be considered: is there evidence of temporality? Strength? Consistency? Specificity? Biological gradient? Plausibility, Coherence? Analogy and experimental? This should be explored in detail in this exploratory study, so a table summarising whether we have evidence to check each of the Hill criteria and the reference could summarise the results

Response: We thank the reviewer for the suggested approach to evaluate the two studies included in this scoping review. As a prior study had highlighted (Munn et al. BMC Med Res Methodol, 2028; 18: 14), a scoping review focuses on mapping the literature and identifying gaps for studies. It also can help generate evidence for systematic reviews. Specifically, Claire Tope (https://www.open.ac.uk/blogs/erc/index.php/2022/01/04/scoping-review-systematic-review-or-review-of-the-literature-what-is-the-difference/#:~:text=A%20systematic%20review%20will%20typically,assess%20quality%20of%20the%20studies)_wrote - A systematic review will typically focus on providing a critically appraised and synthesised account and so may draw on a relatively narrow range of quality assessed studies. A scoping study is likely to draw on a broader range of studies but less likely to assess quality of the studies. The Bradford Hill criteria, first proposed in 1965 by Sir Austin Bradford Hill, provide a framework to determine if one can justifiably move from an observed association to a verdict of causation. A systematic review is best designed to discuss issues the reviewer points to: Strength, Consistency, Specificity, Temporality, Biological gradient, Plausibility, and Coherence, Experiment, and Analogy as criteria for causation. While an evaluation of these attributes is important, they are outside the scope of the current review. 

5. The sentence "High concentrations of natural fluoride in groundwater and soils may cause enamel defects, delayed tooth eruption and endemic skeletal and dental fluorosis [39-44]" needs to be balanced with the fact that caries prevalence is also lower in these areas. See Kut et al. 2016. A review of fluoride in African groundwater and local remediation methods. Groundwater for Sustainable Development 2-3, 190-212.

Response: Thanks for raising this. In lines 309-312, we wrote: Fluoride in water is effective at reducing caries levels in both primary and permanent dentition in children by providing a constant exposure to fluoride ions in the oral cavity [92]. Supra‐optimal levels have been linked to dental fluorosis, and severe dental fluorosis is a risk factor for caries [93].

6. The raw data of potential papers should be included as supplementary material or in a dedicated data research repository.

Response: We have included the search strings for the study which can enable access to the entire database that we searched. The details are included as Supplemental file 1. The details of the two studies included in this study are written up in the table 2. 

Reviewer #2: The research will contribute to the literature. It is a topic that remains current and has undergone a good literature review. Being a multicentric study increases its value. The sources could have been more up-to-date in terms of the year. It is important to highlight the need for more research related to ecological studies.

Response: Thanks for the constructive feedback. We have improved the literature review as highlighted by the reviewer #1. We have also highlighted the need for more ecological studies. We wrote in lines 316-317: Ecological studies may provide preliminary evidence suggesting the plausibility of these links.

Reviewer #3: The manuscript 'A scoping review on the association between early childhood caries and life on land: the Sustainable Development Goal 15' is an excellent and need of the time study.

PRISMA-ScR guidelines are followed well.

Whole manuscript is written in standard English.

Results are meticulously drawn and are well interpreted.

Conclusion and need of further studies are nicely mentioned.

Response: Thanks for the constructive feedback.

---

## [Decision Letter · Decision Letter 1]

7 May 2024

PONE-D-23-43962R1A scoping review on the association between early childhood caries and life on land: the Sustainable Development Goal 15PLOS ONE

Dear Dr. Foláyan,

Thank you for submitting your manuscript to PLOS ONE. After careful consideration, we feel that it has merit but does not fully meet PLOS ONE’s publication criteria as it currently stands. Therefore, we invite you to submit a revised version of the manuscript that addresses the points raised during the review process.

We look forward to receiving your revised manuscript.

Kind regards,

Hadi Ghasemi

Academic Editor

PLOS ONE

Journal Requirements:

Reviewers' comments:

Reviewer's Responses to Questions

**Comments to the Author**

1. If the authors have adequately addressed your comments raised in a previous round of review and you feel that this manuscript is now acceptable for publication, you may indicate that here to bypass the “Comments to the Author” section, enter your conflict of interest statement in the “Confidential to Editor” section, and submit your "Accept" recommendation.

Reviewer #1: All comments have been addressed

2. Is the manuscript technically sound, and do the data support the conclusions?

Reviewer #1: Partly

3. Has the statistical analysis been performed appropriately and rigorously? 

Reviewer #1: N/A

4. Have the authors made all data underlying the findings in their manuscript fully available?

Reviewer #1: Yes

5. Is the manuscript presented in an intelligible fashion and written in standard English?

Reviewer #1: Yes

6. Review Comments to the Author

**Reviewer #1:** I thank the authors for incorporating my previous comments into this manuscript.

To further improve the quality of this document and to ensure that it has any impact, I recommend that the discussion section be expanded to explicitly detail how future research aimed at establishing **or refuting** the association between early childhood caries (ECC) and life on land (Sustainable Development Goal 15) should be designed. It is essential that this section addresses the methodological issues (design? sample size? hypothesis? variables to measure? analysis?) and suggests specific details that such research should include to robustly test the hypothesis of this association. This would provide *clear* guidance for future studies and help advance this research area. This could be included as a detailed table.

7. PLOS authors have the option to publish the peer review history of their article (what does this mean?). If published, this will include your full peer review and any attached files.

Reviewer #1: No

---

## [Author Response · Author response to Decision Letter 1]

10 May 2024

Reviewer #1: I thank the authors for incorporating my previous comments into this manuscript. 

To further improve the quality of this document and to ensure that it has any impact, I recommend that the discussion section be expanded to explicitly detail how future research aimed at establishing **or refuting** the association between early childhood caries (ECC) and life on land (Sustainable Development Goal 15) should be designed. It is essential that this section addresses the methodological issues (design? sample size? hypothesis? variables to measure? analysis?) and suggests specific details that such research should include to robustly test the hypothesis of this association. This would provide *clear* guidance for future studies and help advance this research area. This could be included as a detailed table.

RESPONSE: We thank the Reviewer for the insightful suggestion. At the end of Discussion, we added a new part where we recommended future research directions. We aimed to phrase them as generalized, non exhaustive recommendations to draw the attention of researchers to gaps and sum up the points we raised earlier throughout the paper. We believe that setting an actionable research agenda should follow a systematic approach that draws on the perspectives of multiple stakeholders including researchers, policy makers, practitioners, industry representatives, lay people and others. Thus, we refrained from providing a prescriptive narrative that would overstate our role and the scope of the paper and tried to move beyond specific aims, hypotheses, sample sizes, definition of variables, analytic strategies, and other related details.

---

## [Editor Report · Decision Letter 2]

14 May 2024

A scoping review on the association between early childhood caries and life on land: the Sustainable Development Goal 15

PONE-D-23-43962R2

Dear Dr. Morẹ́nikẹ́ Oluwátóyìn Foláyan,

We’re pleased to inform you that your manuscript has been judged scientifically suitable for publication and will be formally accepted for publication once it meets all outstanding technical requirements.

Kind regards,

Hadi Ghasemi

Academic Editor

PLOS ONE
---

## [Editor Report · Acceptance letter]

23 May 2024

PONE-D-23-43962R2 

PLOS ONE

Dear Dr. Foláyan, 

I'm pleased to inform you that your manuscript has been deemed suitable for publication in PLOS ONE. Congratulations! Your manuscript is now being handed over to our production team.

Kind regards, 

on behalf of

Dr. Hadi Ghasemi 

Academic Editor

PLOS ONE